# Implantable cardioverter-defibrillator therapy after resuscitation from cardiac arrest in vasospastic angina: A retrospective study

**Kazuya Tateishi**[1]\*, **Yusuke Kondo**[1], **Yuichi Saito**[1], **Hideki Kitahara**[1], **Kenichi Fukushima**[2], **Hidehisa Takahashi**[2], **Daichi Yamashita**[1], **Koichi Ohashi**[3], **Ko Suzuki**[3], **Osamu Hashimoto**[4], **Yoshiaki Sakai**[4], **Yoshio Kobayashi**[1]

**1** Department of Cardiovascular Medicine, Chiba University Graduate School of Medicine, Chiba, Chiba, Japan, **2** Department of Cardiovascular Medicine, Matsudo City General Hospital, Matsudo, Chiba, Japan, **3** Department of Cardiology, Tokyo Metropolitan Bokutoh Hospital, Sumida-ku, Tokyo, Japan, **4** Department of Cardiovascular Medicine, Chiba Emergency Medical Center, Chiba, Chiba, Japan

\* kxtateishi@gmail.com

**Data Availability Statement:** All relevant data are within the manuscript and its Supporting Information files.

## Abstract

Patients with vasospastic angina (VSA) who are resuscitated from sudden cardiac arrest (SCA) are at a high risk of recurrent lethal arrhythmia and cardiovascular events. However, the benefit of the implantable cardioverter-defibrillator (ICD) therapy in this population has not been fully elucidated. The present study aimed to analyze the prognostic impact of ICD therapy on patients with VSA and SCA. A total of 280 patients who were resuscitated from SCA and received an ICD for secondary prophylaxis were included in the present multicenter registry. The patients were divided into two groups on the basis of the presence of VSA. The primary endpoint was a composite of all-cause death and appropriate ICD therapy (appropriate anti-tachycardia pacing and shock) for recurrent ventricular arrhythmias. Of 280 patients, 51 (18%) had VSA. Among those without VSA, ischemic cardiomyopathy was the main cause of SCA (38%), followed by non-ischemic cardiomyopathies (18%) and Brugada syndrome (7%). Twenty-three (8%) patients were dead and 72 (26%) received appropriate ICD therapy during a median follow-up period of 3.8 years. There was no significant difference in the incidence of the primary endpoint between patients with and without VSA (24% vs. 33%, p = 0.19). In a cohort of patients who received an ICD for secondary prophylaxis, long-term clinical outcomes were not different between those with VSA and those with other cardiac diseases after SCA, suggesting ICD therapy may be considered in patients with VSA and those with other etiologies who were resuscitated from SCA.

## Introduction

Vasospastic angina (VSA) is a benign disorder [1]; however, it is associated with serious cardiac events, including acute coronary syndrome and life-threatening ventricular arrhythmias [2, 3]. Accumulating evidence indicates that VSA patients with sudden cardiac arrest (SCA) are at a high risk of recurrent lethal ventricular arrhythmias, even if treated with optimal

**Funding:** The authors received no specific funding for this work.

**Competing interests:** The authors have declared that no competing interests exist.

medical therapy [4–6]. In patients resuscitated from SCA, an implantable cardioverter-defi-brillator (ICD) is recommended. ICD therapy as secondary prophylaxis of SCA is a) a Class IIa recommendation for patients with VSA and SCA in whom medical therapy is ineffective or not tolerated and b) a Class IIb recommendation for patients with VSA and SCA in addition to medical therapy [7, 8]. Although recent studies have indicated the beneficial effects of ICD implantation in patients with VSA and SCA [6, 9, 10], previous reports showed no significant effects of ICD therapy or a benign prognosis without ICD therapy [5, 11]. Therefore, the effects of ICD implantation on VSA and SCA remain inconclusive. In this multicenter study, we evaluated the prognostic impact of ICD therapy as secondary prophylaxis in patients with VSA and compared the findings with those for other etiologies of ventricular arrhythmias in a cohort of patients resuscitated from SCA.

## Materials and methods

### Study design and population

Between January 2012 and November 2019, a total of 744 patients underwent ICD implantation at four participating institutions (Tokyo Metropolitan Bokutoh Hospital, Matsudo City General Hospital, Chiba Emergency Medical Center, and Chiba University Hospital) in Japan. After excluding patients aged <18 years and those with an indication for primary prophylaxis, cardiac resynchronization therapy with a defibrillator, or subcutaneous ICD therapy, 286 patients were included in this retrospective study (Fig 1). SCA survivors with VSA who did not receive ICD therapy from January 2012 to November 2019 at the four hospitals were also included as a supplemental cohort (Fig 1).

As subcutaneous ICD cannot provide anti-tachycardia pacing therapy for ventricular tachy-cardia, we excluded patients with subcutaneous ICD. The included patients were further classi-fied into two groups according to the presence or absence of VSA. We obtained the following data from medical records: age, gender, body mass index, comorbidities (hypertension, diabe-tes mellitus, dyslipidemia), family history, laboratory data (hemoglobin, estimated glomerular filtration rate, brain natriuretic peptide), left ventricular ejection fraction, initial rhythm of SCA, cause of SCA, medications at discharge, appropriate and inappropriate ICD therapy, ICD infection/lead disconnection, details of ACh provocation test, survival or death, and cause of death.

### Definitions, diagnosis, and treatment

VSA was diagnosed based on the guidelines for the diagnosis and treatment of patients with VSA given by the Japanese Circulation Society using the following definitions: (1) a positive acetylcholine provocation test, (2) severe coronary stenosis on emergency coronary angiogra-phy that was promptly resolved by injection of nitrates following the patient's arrival, and (3) normal coronary arteries but chest attacks accompanied by transient ST-segment elevation on electrocardiography [12]. In addition, according to the report by Myerburg et al. [2], VSA was determined as a cause of SCA with the following criteria: (1) absence of a previous history of heart diseases, (2) normal left ventricular ejection fraction and no wall motion abnormality, (3) absence of significant coronary artery stenosis (American Heart Association classification ≥75%), and (4) absence of identifiable or reversible causes of lethal ventricular arrhythmias (e.g., electrolyte disturbances, metabolic disturbances, and intoxications/drugs). Patients who were not diagnosed with VSA comprised the non-VSA group. In the present study, VSA was diagnosed using the acetylcholine provocation test in 73% of cases.

Intracoronary acetylcholine provocation tests were performed according to the guidelines [12], as previously reported [13–19]. In brief, after the insertion of a temporary pacing

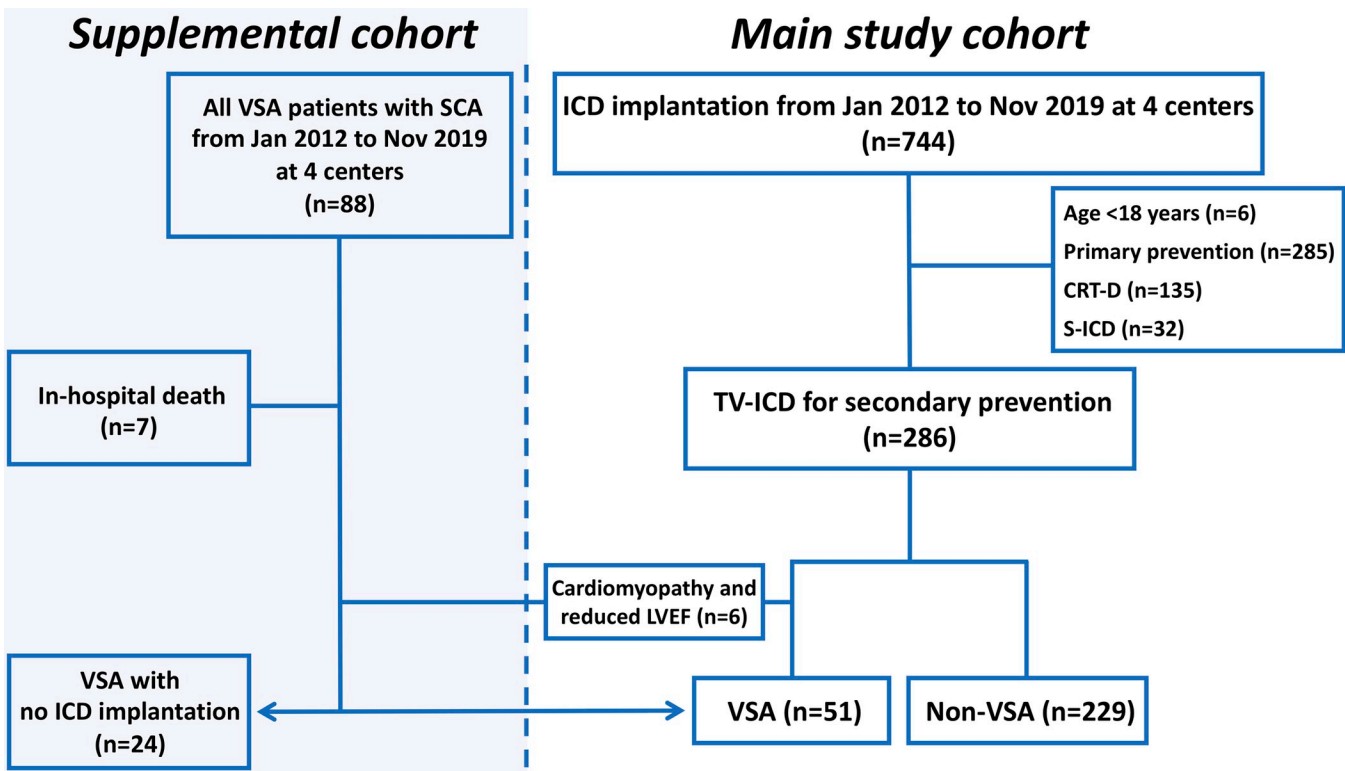

**Fig 1. Flowchart of the study population.** CRT-D, cardiac resynchronization therapy with a defibrillator; LVEF, left ventricular ejection fraction; SCA, sudden cardiac arrest; S-ICD, subcutaneous implantable cardioverter-defibrillator; TV-ICD, transvenous implantable cardioverter-defibrillator; VSA, vasospastic angina.

electrode in the right ventricle, acetylcholine was injected at incremental doses of 20, 50, and 100 μg into the left coronary artery and 20 and 50 μg into the right coronary artery over 20 seconds. Coronary angiography was performed 1 min after the start of each injection. After the acetylcholine provocation test, nitrate was administered into the right and left coronary arteries, and coronary angiography was again performed. Angiographic coronary artery spasm, which was defined as total or subtotal occlusion induced during the acetylcholine provocation test, was assessed by experienced cardiologists who were blinded to the patients' clinical characteristics. A positive acetylcholine provocation test was defined as angiographic coronary artery spasm accompanied by chest pain or ischemic electrocardiographic changes.

All patients underwent a comprehensive non-invasive clinical assessment, including electrocardiography, echocardiography, coronary angiography, cardiac magnetic resonance imaging, and computed tomography. Ischemic cardiomyopathy, non-ischemic cardiomyopathies, Brugada syndrome, long QT syndrome, and cardiac sarcoidosis were diagnosed according to the relevant guidelines [20–22]. Some patients with VSA may have had other overlapping cardiac disorders. Patients with ventricular arrhythmias due to acute myocardial infarction who received primary percutaneous coronary intervention showed no indications for secondary prophylactic ICD therapy according to the guidelines. Therefore, the patients in this study did not have SCA associated with acute coronary syndrome. Post-cardiac arrest care was undertaken based on local standards, such as targeted temperature management and optimal medication. In our institutions, ICD implantation is primarily intended to treat patients with VSA resuscitated from SCA. The ICD was implanted during hospitalization and programmed at the physician's discretion.

## Endpoints

The primary endpoint was a composite of all-cause death and appropriate ICD therapy, including anti-tachycardia pacing and shock for recurrent ventricular arrhythmias. These events were evaluated throughout the follow-up period from the date of ICD implantation. Follow-up data after discharge were obtained by a review of medical records or a telephone interview with the patient or the patient's family members. The clinical outcomes according to ICD implantation were also evaluated in patients with VSA and SCA.

## Statistical analysis

Statistical analysis was performed using the SAS statistical software package version 9.4 (SAS Institute, Cary, NC, USA). Continuous variables are expressed as mean ± standard deviation when normally distributed and as median and interquartile range when non-normally distributed. Categorical data are presented as absolute numbers and percentages. Continuous variables were compared using Student's t-tests or the Mann–Whitney U tests, as appropriate. Categorical variables were compared using the chi-squared test or Fisher's exact test. The cumulative event-free survival rates were calculated using the Kaplan–Meier method and compared using the log-rank test. A Cox proportional-hazards model was used to estimate unadjusted and adjusted hazard ratios with corresponding 95% confidence intervals. Along with VSA, factors associated with the primary endpoint in the univariable analysis ($P < 0.100$) and the left ventricular ejection fraction, a well-known predictor of recurrent SCA [23, 24], were included in the multivariable analysis. A $P$-value of $<0.05$ was considered statistically significant.

## Ethical approval

All procedures performed in studies involving human participants were conducted in accordance with the ethical standards of the Ethical Committee of Chiba University (unique identifier: 3562) and with the 1964 Helsinki Declaration and its later amendments or comparable ethical standards. Since this was a retrospective study, we couldn't obtain written/verbal informed consent from participants for inclusion in the present study. Therefore, the opt-out method, using a poster approved by the Ethical Committee of Chiba University, was utilized to obtain informed consent.

## Results

Among 286 patients resuscitated from SCA who received ICD therapy as secondary prophylaxis, six patients with VSA with reduced left ventricular ejection fraction and cardiomyopathy were excluded. A total of 280 patients were included in this study and divided into two groups: a VSA group (n = 51) and a non-VSA group (n = 229) (Fig 1). Of the patients with VSA, 37 (73%) were diagnosed using the intracoronary acetylcholine provocation test, 13 (25%) by spontaneous ST-segment elevation on ECG, and 1 (2%) by spontaneous coronary vasospasm on emergent coronary angiography. The details in diagnosing VSA are shown in S1 Table. Patients with VSA were significantly younger and had fewer cardiovascular risk factors, such as diabetes and renal impairment, than those without VSA; however, current smoking was more prevalent in patients with VSA than in those without (Table 1). Among patients without VSA, ischemic cardiomyopathy was the leading cause of SCA (38%), followed by non-ischemic cardiomyopathies (18%) and Brugada syndrome (7%) (Table 1). In the VSA group, calcium channel blockers, long-acting nitrate, and nicorandil were used in 94%, 61%, and 57% of patients, respectively. All patients with VSA received at least one oral vasodilator.

**Table 1. Patient characteristics.**

| Variables | All patients (n = 280) | VSA (n = 51) | Non-VSA (n = 229) | P-values |
|---|---|---|---|---|
| Age (years) | 59.9 ± 14.7 | 54.6 ± 11.7 | 61.1 ± 15.1 | 0.004 |
| Male | 219 (78%) | 38 (75%) | 181 (79%) | 0.485 |
| Body mass index (kg/m$^2$) | 23.4 ± 3.9 | 23.1 ± 3.9 | 23.4 ± 3.9 | 0.557 |
| Hypertension | 154 (55%) | 26 (51%) | 128 (56%) | 0.524 |
| Diabetes mellitus | 71 (25%) | 7 (14%) | 64 (28%) | 0.026 |
| Dyslipidaemia | 109 (39%) | 16 (31%) | 93 (41%) | 0.216 |
| Current smoker* | 43/252 (17%) | 14/45 (31%) | 29/207 (14%) | 0.009 |
| Hemoglobin (g/dl) | 13.0 ± 2.0 | 13.4 ± 2.2 | 12.9 ± 2.0 | 0.152 |
| eGFR (ml/min/1.73 m$^2$) | 65.5 ± 24.2 | 72.3 ± 19.0 | 64.0 ± 25.0 | 0.027 |
| BNP (pg/ml) | 110 [42–314] | 53 [15–91] | 135 [49–356] | 0.001 |
| LVEF (%) | 51.8 ± 15.9 | 64.7 ± 7.8 | 48.0 ± 15.6 | <0.001 |
| Family history of SCA | 11 (4%) | 0 (0%) | 11 (5%) | 0.224 |
| Initial rhythm of SCA | | | | 0.226 |
| VT/VF | 275 (98%) | 49 (96%) | 226 (99%) | |
| PEA/Asystole | 5 (2%) | 2 (4%) | 3 (1%) | |
| ACh provocation test | 56 (20%) | 37 (73%) | 19 (8%) | <0.001 |
| Diagnosis | | | | <0.001 |
| VSA | 51 (18%) | 51 (100%) | 0 (0%) | |
| Ischemic cardiomyopathy | 86 (31%) | 0 (0%) | 86 (38%) | |
| Non-ischemic cardiomyopathies | 42 (15%) | 0 (0%) | 42 (18%) | |
| Brugada syndrome | 17 (6%) | 1 (2%) | 16 (7%) | |
| Long QT syndrome | 14 (5%) | 2 (4%) | 12 (5%) | |
| Sarcoidosis | 13 (5%) | 0 (0%) | 13 (6%) | |
| Others | 60 (21%) | 0 (0%) | 60 (26%) | |
| Medication | | | | |
| Calcium channel blocker | 101 (36%) | 48 (94%) | 53 (23%) | <0.001 |
| Long-acting nitrate | 33 (12%) | 31 (61%) | 2 (1%) | <0.001 |
| Nicorandil | 48 (17%) | 29 (57%) | 19 (8%) | <0.001 |
| ACE-I or ARB | 132 (47%) | 5 (10%) | 127 (55%) | <0.001 |
| β-blocker | 183 (65%) | 4 (8%) | 179 (78%) | <0.001 |
| Amiodarone | 111 (40%) | 5 (10%) | 106 (46%) | <0.001 |
| Statin | 104 (37%) | 7 (14%) | 97 (42%) | <0.001 |

Data are shown as mean ± standard deviation, median [interquartile range], or number (%). * Data for current smoking were missing in 28 patients. ACE-I, angiotensin converting enzyme inhibitor; ACh, acetylcholine; ARB, angiotensin receptor blockers; BNP, brain natriuretic peptide; eGFR, estimated glomerular filtration rate; LVEF, left ventricular ejection fraction; PEA, pulseless electrical activity; SCA, sudden cardiac arrest; VF, ventricular fibrillation; VSA, vasospastic angina; VT, ventricular tachycardia.

During a median follow-up period of 3.8 (2.1–5.6) years, 24 (8%) patients died and 72 (26%) received appropriate ICD therapy (Table 2). In the VSA group, 11 (22%) patients had undergone ICD therapy (7 patients received shock and 4 anti-tachycardia pacing only), and further examinations on re-admission did not reveal another reason for the recurrent, lethal ventricular arrhythmias. There was no significant difference in the composite endpoint during the study period between the VSA and non-VSA groups (24% vs. 33%, p = 0.19). The causes of death are shown in Table 3. One patient (2%) in the VSA group died of a cardiac cause compared to 10 patients (4%) in the non-VSA group. The Kaplan–Meier analysis demonstrated that all-cause death rates and appropriate ICD therapy were not significantly different between

**Table 2. Clinical outcomes.**

| Variables | All patients (n = 280) | VSA (n = 51) | Non-VSA (n = 229) | P-values |
|---|---|---|---|---|
| Follow-up duration (years) | 3.8 [2.1–5.6] | 4.1 [2.0–6.9] | 3.8 [2.1–5.3] | 0.196 |
| Composite endpoint | 87 (31%) | 12 (24%) | 75 (33%) | 0.189 |
| All-cause death | 24 (8%) | 2 (4%) | 22 (10%) | 0.271 |
| Cardiac death | 11 (4%) | 1 (2%) | 10 (4%) | 0.695 |
| Appropriate ICD therapy | 72 (26%) | 11 (22%) | 61 (27%) | 0.595 |
| ATP only | 25 (9%) | 4 (8%) | 21 (9%) | 1.000 |
| Shock | 47 (17%) | 7 (14%) | 40 (17%) | 0.679 |
| Inappropriate ICD therapy | 23 (8%) | 5 (10%) | 18 (8%) | 0.583 |
| Infection/lead disconnection | 5 (2%) | 2 (4%) | 3 (1%) | 0.226 |

Data are shown as median [interquartile range] or number (%). ATP, anti-tachycardia pacing; ICD, implantable cardioverter-defibrillator; VSA, vasospastic angina

patients with and without VSA in the main study cohort (Fig 2). Although patient characteristics differed widely according to etiology, the incidence of the primary endpoint was not significantly different (S2 Table, Fig 3). Multivariable analysis showed that VSA was not associated with all-cause death and appropriate ICD therapy (Table 4). We also analyzed predictors of

**Table 3. Clinical data of patients who died during the follow-up period.**

| Patient # | Age | Sex | Diagnosis | Follow-up (years) | Appropriate ICD shocks | Cardiac death | Cause of death |
|---|---|---|---|---|---|---|---|
| 1 | 76 | Male | VSA | 1.0 | No | No | Cancer |
| 2 | 78 | Female | VSA | 1.5 | No | Yes | SCA |
| 3 | 65 | Male | ICM | 0.6 | No | Yes | CHF |
| 4 | 71 | Female | Non-ICM | 2.5 | No | Yes | CHF |
| 5 | 70 | Male | Non-ICM | 4.1 | Yes | Yes | SCA |
| 6 | 64 | Male | Sarcoidosis | 3.1 | No | Yes | CHF |
| 7 | 59 | Male | Others | 2.5 | No | No | Stroke |
| 8 | 73 | Female | Sarcoidosis | 0.9 | No | No | Cancer |
| 9 | 72 | Male | Others | 3.5 | No | Yes | SCA |
| 10 | 50 | Female | ICM | 0.5 | No | Yes | CHF |
| 11 | 78 | Male | ICM | 2.6 | No | No | Cancer |
| 12 | 76 | Male | LQTS | 0.8 | No | No | Cancer |
| 13 | 49 | Male | ICM | 0.4 | No | No | Trauma |
| 14 | 66 | Male | Others | 1.9 | No | No | Cancer |
| 15 | 74 | Male | Others | 0.7 | No | No | Infection |
| 16 | 76 | Male | Sarcoidosis | 4.0 | Yes | Yes | CHF |
| 17 | 65 | Male | ICM | 4.7 | Yes | Yes | CHF |
| 18 | 65 | Male | Non-ICM | 1.8 | Yes | No | Infection |
| 19 | 82 | Male | ICM | 1.0 | No | No | Infection |
| 20 | 88 | Male | ICM | 3.9 | No | No | Cancer |
| 21 | 74 | Male | Others | 5.2 | Yes | Yes | SCA |
| 22 | 83 | Male | Others | 1.4 | No | No | Stroke |
| 23 | 72 | Female | Non-ICM | 1.6 | No | Yes | CHF |
| 24 | 83 | Male | Non-ICM | 0.5 | Yes | No | Infection |

CHF, congestive heart failure; ICD, implantable cardioverter defibrillator; ICM, ischemic cardiomyopathy; LQTS, long QT syndrome; SCA, sudden cardiac arrest; VSA, vasospastic angina

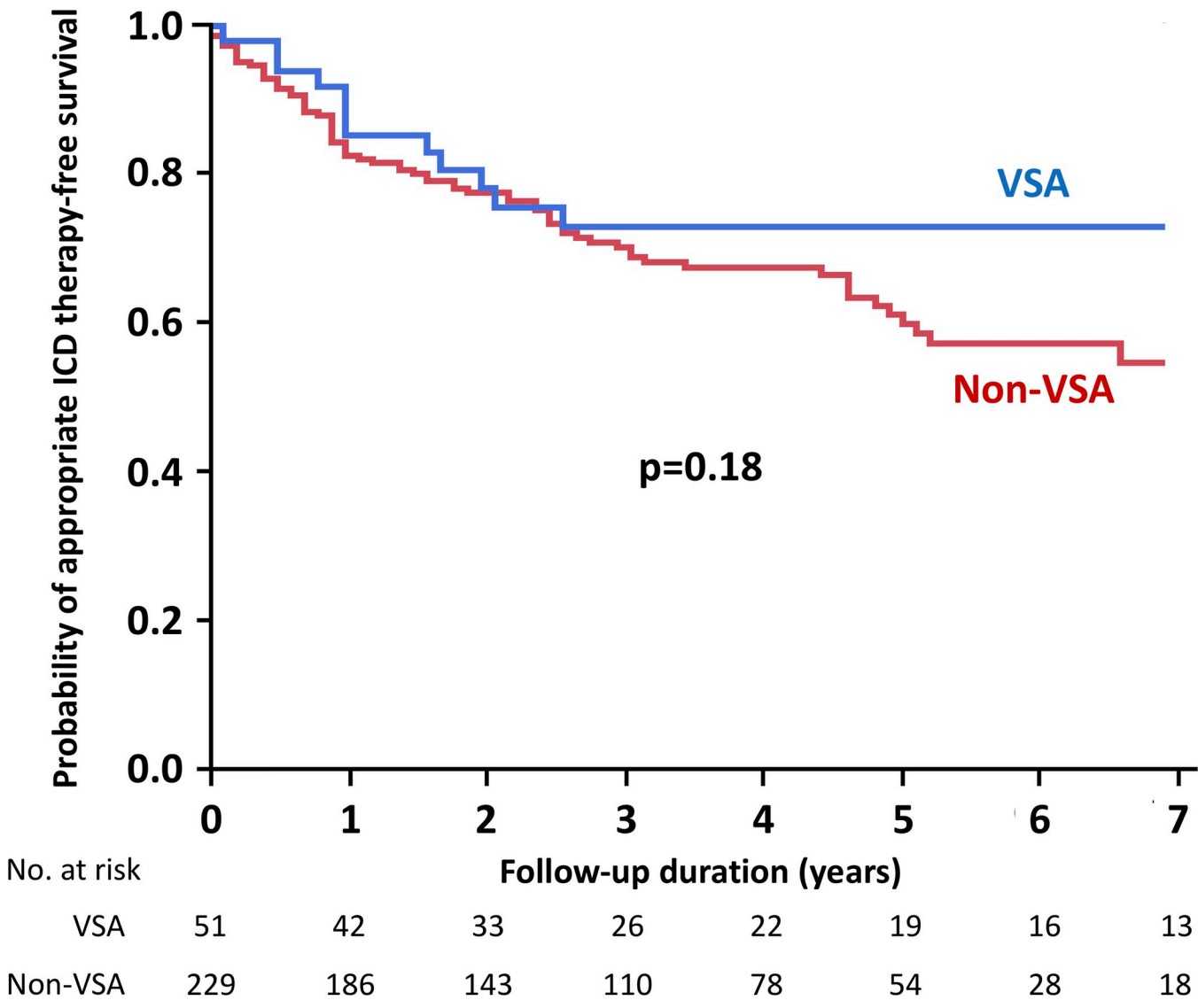

**Fig 2. Kaplan–Meier curves for the probability of appropriate ICD therapy-free survival.** ICD, implantable cardioverter-defibrillator; VSA, vasospastic angina.

all-cause death and appropriate ICD therapy in patients with VSA, but no significant factors were found (S3 Table).

In the comparison of patients with VSA according to ICD implantation, 24 of 75 (32%) patients did not receive ICD after SCA (Fig 1). Although there were notable differences in patient characteristics according to ICD implantation after SCA (S4 Table), Kaplan–Meier curve analysis showed a lower rate of cardiac death in patients with ICD than in those without (S1 Fig).

## Discussion

The main findings of this multicenter study were as follows: (1) VSA was found in approximately 20% of patients resuscitated from SCA who received ICD therapy as secondary prophylaxis, and (2) long-term clinical outcomes were not different between patients with and

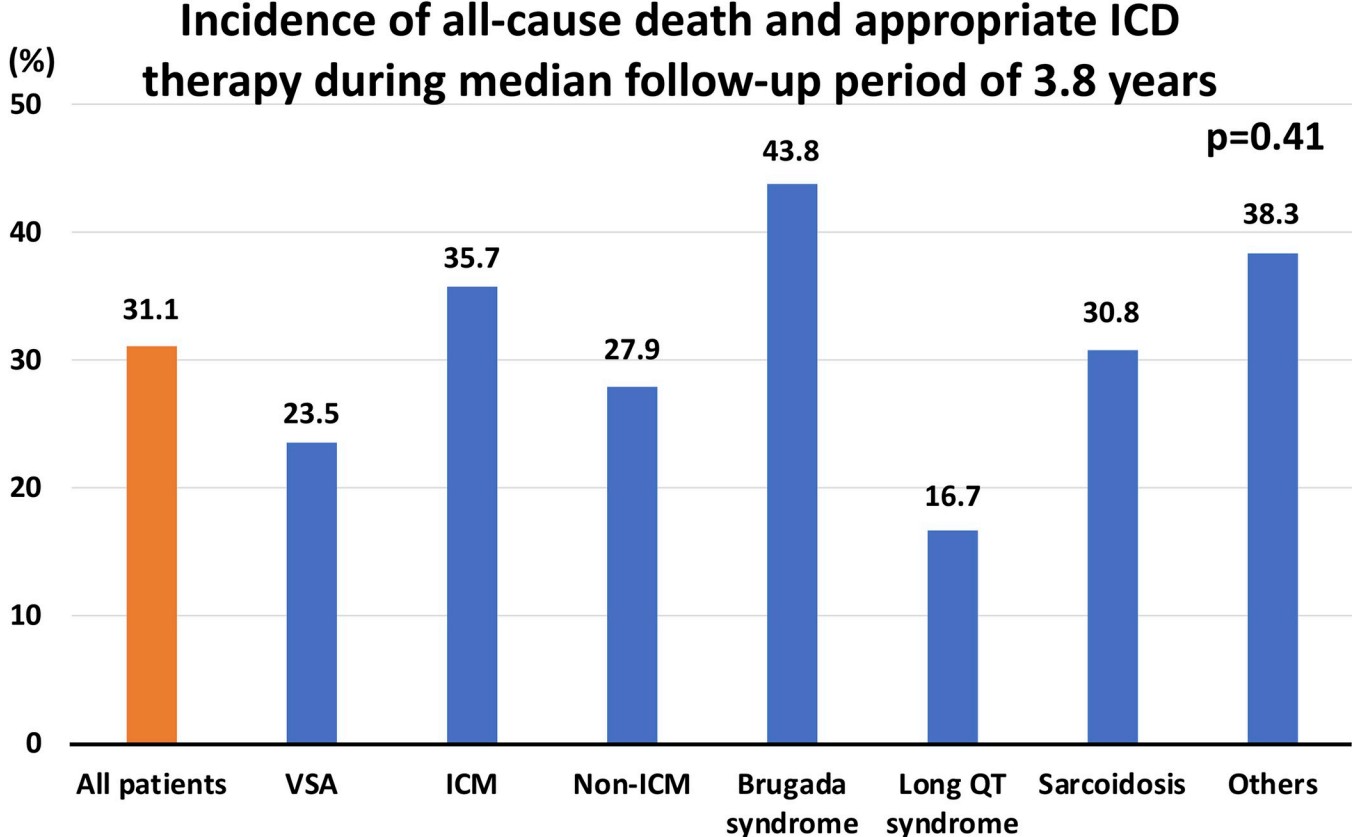

**Fig 3. Incidence of all-cause death and appropriate ICD therapy among etiologies.** ICD, implantable cardioverter-defibrillator; VSA, vasospastic angina; ICM, ischemic cardiomyopathy.

without VSA across a broad range of SCA etiologies in this patient population. To the best of our knowledge, this is the largest report to have investigated the prognosis of patients with VSA who received ICD implantation as secondary prophylaxis.

## VSA and SCA

VSA has been recognized as a benign disorder, given its prognosis [1]. However, coronary spasms occasionally play an important role in ventricular arrhythmia. Recent studies have demonstrated that VSA accounts for 8%–11% of out-of-hospital cardiac arrests [25, 26], and a Japanese multicenter registry showed that out-of-hospital cardiac arrest is the strongest predictor of future major adverse events in patients with VSA [4]. This study found that VSA accounted for 18% of SCA etiologies in a secondary prophylactic population. Notably, VSA in this study was diagnosed using an intracoronary acetylcholine provocation test in 73% of the patients, reinforcing the importance of this invasive technique in examining the cause of SCA. As VSA requires specialized treatment strategies (e.g., calcium channel blockers) [25], it can be assumed that its diagnosis may contribute to improved clinical outcomes. In fact, the incidence of the primary endpoint was lower in patients with VSA than in those with other etiologies, such as Brugada syndrome and other cardiac disorders (mainly, idiopathic ventricular fibrillation), for which no specialized medical treatment options exist. Although we previously showed the safety and usefulness of the acetylcholine provocation test in an acute coronary syndrome setting [16], its feasibility in SCA has not been established. We believe that invasive

**Table 4. Factors associated with death and recurrent ventricular arrhythmias.**

| Variables | Univariable analysis | | | Multivariable analysis | | |
|---|---|---|---|---|---|---|
| | HR | *P*-values | 95% CI | HR | *P*-values | 95% CI |
| Age (years) | 1.01 | 0.436 | 0.99–1.02 | | | |
| Male | 1.07 | 0.783 | 0.65–1.79 | | | |
| Body mass index (kg/m$^2$) | 0.96 | 0.171 | 0.90–1.02 | | | |
| Hypertension | 0.76 | 0.204 | 0.50–1.16 | | | |
| Dyslipidaemia | 1.12 | 0.607 | 0.73–1.71 | | | |
| Diabetes mellitus | 0.95 | 0.838 | 0.59–1.54 | | | |
| Chronic kidney disease | 1.73 | 0.011 | 1.13–2.65 | 1.87 | 0.013 | 1.14–3.07 |
| Family history of SCA | 0.44 | 0.253 | 0.11–1.80 | | | |
| Left ventricular ejection fraction (%) | 0.99 | 0.156 | 0.98–1.00 | 0.99 | 0.396 | 0.98–1.01 |
| Diagnosis | | | | | | |
| VSA | 0.66 | 0.168 | 0.36–1.22 | 0.84 | 0.644 | 0.41–1.73 |
| ICM | 0.89 | 0.631 | 0.56–1.43 | | | |
| Non-ICMs | 1.08 | 0.792 | 0.62–1.88 | | | |
| Brugada syndrome | 1.87 | 0.092 | 0.90–3.87 | 2.46 | 0.075 | 0.91–6.62 |
| Long QT syndrome | 0.48 | 0.210 | 0.15–1.52 | | | |
| Sarcoidosis | 0.91 | 0.846 | 0.33–2.47 | | | |
| Others | 1.49 | 0.100 | 0.93–2.41 | | | |

Chronic kidney disease was defined as glomerular filtration rate < 60 mL/min/1.73 m$^2$. CI, confidence interval; HR, hazard ratio; ICM, ischemic cardiomyopathy; SCA, sudden cardiac arrest; VSA, vasospastic angina

diagnostic strategies, including the acetylcholine test, are essential for SCA diagnosis, as recommended by an expert consensus document [27].

## Prognostic impact of VSA on patients with SCA

A few reports have suggested no clear benefit of ICD or a benign prognosis after SCA on patients with VSA [5, 11]. For instance, Yamashina et al. followed up 16 patients with VSA and SCA treated with optimal medical therapy and smoking cessation. They found that none of the patients developed recurrent ventricular arrhythmias during the mean follow-up of 67 months [11]. However, the number of patients included in the study was limited. A larger European retrospective survey indicated that 12 of 44 (27%) patients with life-threatening ventricular arrhythmia due to VSA who received ICD therapy experienced recurrent ventricular arrhythmias during the median follow-up period of 59 months [6]. Similarly, an observational study showed that 23.5% of VSA patients with an ICD received appropriate shock therapy during a mean follow-up of 5.5 years [28]; these findings align with our results. Importantly, even after optimal medical treatment, 4 of 12 (33%) patients underwent further appropriate ICD therapy in the European retrospective survey [6], suggesting that ICD implantation may be considered even if a patient receives optimal medical therapy.

Our study demonstrates that more than 20% of patients with VSA, as well as those with other etiologies of SCA, had all-cause death and recurrent ventricular arrhythmias during a follow-up period of 3.8 years despite optimal medical treatment. ICD therapy as secondary prophylaxis of SCA is a Class IIa or IIb recommendation based on the effect of medical treatment in patients with VSA and SCA [7], but most patients with VSA and SCA are reportedly asymptomatic (approximately 75%) [25, 29]. Accordingly, an exact evaluation of the effectiveness of medical treatment for VSA is difficult. ICD therapy may be considered in patients with

VSA and those with other cardiac etiologies after resuscitation from SCA; however, further studies are needed to establish the efficacy of ICD and factors associated with its efficacy in this specific cohort.

## Study limitations

This study has several limitations. First, this is a retrospective study, and the number of included patients was limited. However, this study has one of the largest cohorts of patients with VSA who were resuscitated from SCA. Second, the diagnostic procedures were left to the discretion of physicians, and the acetylcholine provocation test was more frequently performed in this study than in previous studies [6, 30], while only 8.3% of the non-VSA group underwent the acetylcholine provocation test. In addition, the number of patients resuscitated from SCA who were diagnosed with VSA based on the acetylcholine provocation test might have been overestimated because the endothelial function is impaired in patients with cardiac arrest [31]. Therefore, further prospective studies will be needed to investigate the precise prevalence of VSA in patients resuscitated from SCA. Finally, this study was not a randomized controlled trial, and clinical outcomes in patients with VSA and SCA were evaluated in a limited cohort of patients who received ICD implantation as secondary prophylaxis. Therefore, this study did not directly address the effectiveness of ICD in patients with VSA.

## Conclusion

VSA was found in 18% of patients who underwent ICD implantation as secondary prophylaxis, suggesting that intensive strategies may be needed to diagnose VSA. Long-term clinical outcomes were not different between patients with and without VSA across a broad spectrum of SCA etiologies. ICD therapy may be considered in patients with VSA and those with other etiologies after resuscitation from SCA.

## Supporting information

**S1 Table. Diagnostic methods for VSA used for patients with VSA in this study.** Data are shown as the number (%). Ach, acetylcholine; CAG, coronary angiography; ECG, electrocardiogram; LAD, left anterior descending artery; LCX, left circumflex artery; RCA, right coronary artery; VSA, vasospastic angina.
(DOC)

**S2 Table. Patient characteristics according to etiology.** Data are shown as mean ± standard deviation, median [interquartile range], or number (%). * Data for current smoking were missing in 22 patients. ATP, anti-tachycardia pacing; BNP, brain natriuretic peptide; eGFR, estimated glomerular filtration rate; ICD, implantable cardioverter-defibrillator; ICM, ischemic cardiomyopathy; LVEF, left ventricular ejection fraction; PEA, pulseless electrical activity; SCA, sudden cardiac arrest; VF, ventricular fibrillation; VSA, vasospastic angina; VT, ventricular tachycardia.
(DOC)

**S3 Table. Factors associated with death and recurrent ventricular arrhythmias in patients with VSA.** Chronic kidney disease was defined as glomerular filtration rate <60 mL/min/1.73 m$^2$. ACE-I, angiotensin converting enzyme inhibitor; ARB, angiotensin receptor blockers; CI, confidence interval; HR, hazard ratio; SCA, sudden cardiac arrest; VSA, vasospastic angina.
(DOC)

**S4 Table. Characteristics of patients with VSA and SCA according to ICD implantation.**
Data are shown as mean ± standard deviation, median (interquartile range), or number (%). *
Data for current smoking are missing in 7 patients. ACE-I, angiotensin converting enzyme
inhibitor; ACh: acetylcholine; ARB, angiotensin receptor blockers; BNP: brain natriuretic pep-
tide; CAG, coronary angiography; ECG, electrocardiogram; eGFR: estimated glomerular filtra-
tion rate; ICD, implantable cardioverter defibrillator; LAD, left anterior descending artery;
LCX, left circumflex artery; LVEF: left ventricular ejection fraction; PEA: pulseless electrical
activity; RCA, right coronary artery; SCA: sudden cardiac arrest; VF: ventricular fibrillation;
VSA: vasospastic angina; VT: ventricular tachycardia.
(DOCX)

**S1 Fig. Kaplan–Meier curves for the probability of cardiac death-free survival among
patients with VSA and SCA according to ICD implantation.** ICD, implantable cardioverter-
defibrillator; SCA, sudden cardiac arrest; VSA, vasospastic angina.
(TIF)

**S1 Dataset.**
(XLSX)

## Acknowledgments

We thank all hospital staff who assisted in the data collection.

## Author Contributions

**Conceptualization:** Kazuya Tateishi, Yusuke Kondo, Yuichi Saito, Hideki Kitahara.

**Data curation:** Kenichi Fukushima, Hidehisa Takahashi, Daichi Yamashita, Koichi Ohashi,
Ko Suzuki, Osamu Hashimoto, Yoshiaki Sakai.

**Formal analysis:** Kazuya Tateishi.

**Supervision:** Yoshio Kobayashi.

**Writing – original draft:** Kazuya Tateishi.

**Writing – review & editing:** Yusuke Kondo, Yuichi Saito, Hideki Kitahara, Yoshio Kobayashi.

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
