## [Decision Letter · Decision Letter 0]

4 Sep 2022

PONE-D-22-17097Implantable cardioverter-defibrillator therapy after resuscitation from cardiac arrest in vasospastic angina: a retrospective studyPLOS ONE

Dear Dr. Tateishi,

Thank you for submitting your manuscript to PLOS ONE. After careful consideration, we feel that it has merit but does not fully meet PLOS ONE’s publication criteria as it currently stands. Therefore, we invite you to submit a revised version of the manuscript that addresses the points raised during the review process.

We look forward to receiving your revised manuscript.

Kind regards,

Shukri AlSaif

Academic Editor

PLOS ONE

Journal Requirements:

4. Thank you for submitting the above manuscript to PLOS ONE. During our internal evaluation of the manuscript, we found significant text overlap between your submission and the following previously published works, some of which you are an author.

- https://academic.oup.com/eurheartj/article/42/Supplement_1/ehab724.0690/6391837?login=false

Please revise the manuscript to rephrase the duplicated text, cite your sources, and provide details as to how the current manuscript advances on previous work. Please note that further consideration is dependent on the submission of a manuscript that addresses these concerns about the overlap in text with published work.

Reviewers' comments:

Reviewer's Responses to Questions

**Comments to the Author**

1. Is the manuscript technically sound, and do the data support the conclusions?

Reviewer #1: Partly

Reviewer #2: Yes

2. Has the statistical analysis been performed appropriately and rigorously? 

Reviewer #1: Yes

Reviewer #2: Yes

3. Have the authors made all data underlying the findings in their manuscript fully available?

Reviewer #1: Yes

Reviewer #2: Yes

4. Is the manuscript presented in an intelligible fashion and written in standard English?

Reviewer #1: No

Reviewer #2: Yes

5. Review Comments to the Author

Reviewer #1: The paper by Kazuya Tateishi et al. relies on the prognosis of patients with variant angina and sudden cardiac death. They compare implanted patients with VSA with other ICD patients implanted after SCD.

Major comments :

- VSA group (n=51) and non-VSA group (n=229) is rather intriguing since VSA are more excpetional in Europe and 229 non VSA ICD patients over a rather long time is not a so important number of implanted patienst. Could the authors comment on that ?

- the results are mainly presented as tables. A more detailed presentation in the text is probably more readable

- details about the appropriate ICD therapies in the VSA group are mandatory. Did VT/VF occur in relation with other vasospastic events ? or with acute coornary trhombosis ? How many VF and VT ? did some patients experience several events ? what was the anti arrhythmic drug regimen at the time of the recurrence and therafter ?

- where are the comparisons between VSA patients and subtypes of other ICD patients ? (seen in figure 3 but anaysis should be provided)

- Factors associated with death and recurrent ventricular arrhythmias should be separately analyzed for VSA patients

- A better comarison would be with a third group of VSA with cardiac arrest but non implanted. Did the authors have such group of patients to be studied ?

Reviewer #2: Well written paper that acknowledges the limitations involved in a retrospective analysis and is soundly written. Furthermore, the paper highlights a possibly under-diagnosed representation of VSA amongst patients that have survived a sudden cardiac arrest.

6. PLOS authors have the option to publish the peer review history of their article (what does this mean?). If published, this will include your full peer review and any attached files.

Reviewer #1: No

Reviewer #2: No

---

## [Author Response · Author response to Decision Letter 0]

11 Oct 2022

Responses to the comments of Reviewer #1

We appreciate the careful and comprehensive review of our manuscript. In response to the Reviewer’s comments and recommendations, we have answered the questions in a point-by-point fashion and revised our manuscript as follows. The changes we made are shown in red font in the revised manuscript.

1. VSA group (n=51) and non-VSA group (n=229) is rather intriguing since VSA are more exceptional in Europe and 229 non VSA ICD patients over a rather long time is not a so important number of implanted patient. Could the authors comment on that?

As the Reviewer pointed out, the patients with VSA accounted for 18% of the entire population (patients who received ICD implantation for secondary prophylaxis) in the present study. This high prevalence was probably because we are encouraged to perform ACh provocation tests for patients with cardiac arrest of an unknown etiology at our institutions. However, since this is a retrospective study, further prospective studies will be needed to investigate the exact prevalence of VSA in patients with cardiac arrest. We have accordingly added the following sentence to the Study limitations section of the manuscript: 

“Therefore, further prospective studies will be needed to investigate the precise prevalence of VSA in patients resuscitated from SCA.” (page 20, lines 6–7)

2. The results are mainly presented as tables. A more detailed presentation in the text is probably more readable.

We appreciate the Reviewer’s meaningful comment. We have accordingly revised the Results section as follows:

“Of the patients with VSA, 37 (73%) were diagnosed using the intracoronary acetylcholine provocation test, 13 (25%) by spontaneous ST-segment elevation on ECG, and 1 (2%) by spontaneous coronary vasospasm on emergent coronary angiography.” (page 10, lines 10–13).

“Among patients without VSA, ischemic cardiomyopathy was the leading cause of SCA (38%), followed by non-ischemic cardiomyopathies (18%) and Brugada syndrome (7%) (Table 1).” (page 10, lines 16–18)

“There was no significant difference in the composite endpoint during the study period between the VSA and non-VSA groups (24% vs. 33%, p=0.19).” (page 12, line 12 to page 13, line 1)

“One patient (2%) in the VSA group died of a cardiac cause compared to 10 patients (4%) in the non-VSA group.” (page 13, lines 2–3)

3. Details about the appropriate ICD therapies in the VSA group are mandatory. Did VT/VF occur in relation with other vasospastic events? or with acute coronary thrombosis? How many VF and VT? did some patients experience several events? what was the anti-arrhythmic drug regimen at the time of the recurrence and thereafter?

Of the 11 patients in the VSA group who received appropriate ICD therapy, 5 had VF and 6 had sustained VT. Although coronary angiography was not routinely performed for patients without ST-segment elevation who had undergone appropriate ICD therapy, there were no patients with acute coronary thrombosis in this cohort. Since the routine check-up and further examinations performed on re-admission could not reveal any other reason for the recurrent VT/VF in these patients, we believe that coronary vasospasm is one of the major causes of recurrent lethal arrhythmia. 

Actually, 2 patients in the VSA group had undergone several appropriate ICD therapies in the present study. For the patients with VSA who had undergone appropriate ICD therapy, we usually added CCBs (both dihydropyridine and non-dihydropyridine) and long-active nitrate and subsequently escalated them up to the maximum dose. However, since we discontinued follow-up when the first event occurred in this study, we could not obtain detailed information about the changes in anti-arrhythmic medication in these 2 patients. As this point is clinically important, further studies are needed. Therefore, we have added the following sentence to the revised Results section: 

“In the VSA group, 11 (22%) patients had undergone ICD therapy (7 patients received shock and 4 anti-tachycardia pacing only), and further examinations on re-admission did not reveal another reason for the recurrent, lethal ventricular arrhythmias.” (page 12, lines 9–12)

4. Where are the comparisons between VSA patients and subtypes of other ICD patients? (seen in figure 3 but analysis should be provided)

We have added the patient characteristics and outcomes according to each etiology to Table S2. Although patient characteristics differed notably among the groups, there was no significant difference in any outcomes. Thus, we have revised the Results section as follows: 

“Although patient characteristics differed widely according to etiology, the incidence of the primary endpoint was not significantly different (Table S2, Fig 3).” (page 13, lines 5–7)

5. Factors associated with death and recurrent ventricular arrhythmias should be separately analyzed for VSA patients.

As the Reviewer suggested, we have analyzed some factors associated with the primary endpoint (all-cause death and appropriate ICD therapy) in patients with VSA (n=51). However, probably because of the limited sample size, no significant predictors were found (Table S3). We have accordingly added the following sentence to the Results: 

“We also analyzed predictors of all-cause death and appropriate ICD therapy in patients with VSA, but no significant factors were found (Table S3).” (page 13, lines 8–10)

6. A better comparison would be with a third group of VSA with cardiac arrest but non implanted. Did the authors have such group of patients to be studied? 

According to the Reviewer’s suggestion, we have added an analysis of a supplemental cohort, which includes SCA survivors with VSA who did not receive ICD therapy at the four hospitals between January 2012 and November 2019, to the revised manuscript, in addition to the main study cohort (Figure 1). There were 88 SCA survivors with VSA managed during this period at the participating hospitals. After excluding patients who died during the index hospitalization and those with cardiomyopathy or reduced left ventricular ejection fraction, 24 patients were identified as the supplemental cohort. The characteristics of VSA patients with or without ICD therapy are shown in Table S4. Kaplan–Meier curve analysis showed a lower rate of cardiac death in patients with ICD than in those without during the follow-up period (Figure S1); however, the patients’ baseline characteristics differed notably between the two groups (Table S4). We have added Figure 1 and S1, Table S4, and have revised the Methods and Results accordingly. 

“SCA survivors with VSA who did not receive ICD therapy from January 2012 to November 2019 at the four hospitals were also included as a supplemental cohort (Fig 1).” (page 5, lines 9–11) 

“In the comparison of patients with VSA according to ICD implantation, 24 of 75 (32%) patients did not receive ICD after SCA (Fig 1). Although there were notable differences in patient characteristics according to ICD implantation after SCA (Table S4), Kaplan–Meier curve analysis showed a lower rate of cardiac death in patients with ICD than in those without (Fig S1).” (page 16, lines 11–14)

 

Responses to the comments of Reviewer #2

Thank you for your time and input. We sincerely appreciate your comprehensive comments.

---

## [Editor Report · Decision Letter 1]

19 Oct 2022

Implantable cardioverter-defibrillator therapy after resuscitation from cardiac arrest in vasospastic angina: a retrospective study

PONE-D-22-17097R1

Dear Dr. Tateishi,

We’re pleased to inform you that your manuscript has been judged scientifically suitable for publication and will be formally accepted for publication once it meets all outstanding technical requirements.

Kind regards,

Shukri AlSaif

Academic Editor

PLOS ONE
---

## [Editor Report · Acceptance letter]

21 Oct 2022

PONE-D-22-17097R1 

Implantable cardioverter-defibrillator therapy after resuscitation from cardiac arrest in vasospastic angina: a retrospective study 

Dear Dr. Tateishi:

I'm pleased to inform you that your manuscript has been deemed suitable for publication in PLOS ONE. Congratulations! Your manuscript is now with our production department. 

Kind regards, 

on behalf of

Dr. Shukri AlSaif 

Academic Editor

PLOS ONE